# Zinc and Central Nervous System Disorders

**DOI:** 10.3390/nu15092140

**Published:** 2023-04-29

**Authors:** Bangqi Wang, Tianshu Fang, Hongping Chen

**Affiliations:** 1Department of Histology and Embryology, Medical College, Nanchang University, Nanchang 330006, China; 4217119061@email.ncu.edu.cn (B.W.); 4217119068@email.ncu.edu.cn (T.F.); 2Queen Mary School, Medical College, Nanchang University, Nanchang 330006, China

**Keywords:** zinc, central nervous system, neurodegeneration, Alzheimer’s disease, depression, Parkinson’s disease, multiple sclerosis

## Abstract

Zinc (Zn^2+^) is the second most abundant necessary trace element in the human body, exerting a critical role in many physiological processes such as cellular proliferation, transcription, apoptosis, growth, immunity, and wound healing. It is an essential catalyst ion for many enzymes and transcription factors. The maintenance of Zn^2+^ homeostasis is essential for the central nervous system, in which Zn^2+^ is abundantly distributed and accumulates in presynaptic vesicles. Synaptic Zn^2+^ is necessary for neural transmission, playing a pivotal role in neurogenesis, cognition, memory, and learning. Emerging data suggest that disruption of Zn^2+^ homeostasis is associated with several central nervous system disorders including Alzheimer’s disease, depression, Parkinson’s disease, multiple sclerosis, schizophrenia, epilepsy, and traumatic brain injury. Here, we reviewed the correlation between Zn^2+^ and these central nervous system disorders. The potential mechanisms were also included. We hope that this review can provide new clues for the prevention and treatment of nervous system disorders.

## 1. Introduction

Zinc (Zn^2+^), the second most abundant necessary trace element in the human body, is an indispensable co-factor of more than 300 enzymes and 2000 transcription factors. Evidence suggests that about 10% of all proteins need to bind to Zn^2+^ to function properly and their expression levels are also regulated by Zn^2+^ [1]. It plays an important role in many physiological processes, including DNA repair, transcription, protein synthesis, apoptosis, proliferation, wound healing, and immune response [2,3]. Zn^2+^ was found to be involved in rapid ligand exchange reactions as a second messenger in several signal transduction pathways like calcium [4] (Figure 1). Thus, Zn^2+^ deficiency can lead to many disorders, including retarded growth and brain development, immune dysfunction, delayed wound healing, learning disabilities, and olfactory and taste disorders [5,6,7,8].

In normal conditions, Zn^2+^ content in the human body should be kept around 2–3 grams, less than 50 mg/kg, and over 90% of Zn^2+^ exists in cells of bones and muscles [9,10]. The absorption of Zn^2+^ is determined by intake rather than the host’s Zn^2+^ status [11]. Nowadays, Zn^2+^ deficiency remains a global public health problem to be addressed. According to World Health Organization (WHO), 0.8 million (1.4%) deaths worldwide are due to Zn^2+^ deficiency [12], thus appropriate Zn^2+^ supplementation is vital for the prevention and treatment of certain diseases caused by Zn^2+^ deficiency [13]. In addition to the exogenous supplementation, Zn^2+^ also needs to be tightly regulated at a physiological homeostatic level in vivo by a variety of proteins involving Zn^2+^ transporters and binders, especially in the central nervous system (CNS) [14,15]. The maintenance of Zn^2+^ homeostasis is significant for the normal functions of CNS [16]. Given the crucial role of Zn^2+^ in CNS, the disruption of Zn^2+^ homeostasis is correlated to several CNS diseases as a contributing factor [17,18]. The relationship between Zn^2+^ and CNS disorders is complex. Although there is extensive evidence on this topic, the results are conflicting. This review provides an overview of the latest preclinical and clinical data on the role of Zn^2+^ in the pathophysiology of seven central nervous system disorders, aiming to draw a link between altered extracellular/intracellular Zn^2+^ levels and targets which can be influenced by Zn^2+^ homeostasis. Also, the potential of Zn^2+^-based intervention as a therapeutic strategy for these disorders was also discussed in this review.

## 2. Zn^2+^ in the CNS

Zn^2+^ is a significant element required by CNS throughout life. The brain is the organ with the highest Zn^2+^ concentration of about 150 µmol/L in the human body, in which Zn^2+^ accumulates in the cerebral cortex, olfactory cortex, thalamus, hippocampus and amygdala [19]. It plays a critical role in learning, memory, synaptic plasticity, and neurogenesis [20].

Most of the brain Zn^2+^ (80–95%) binds to proteins including Zn^2+^ metalloenzymes and metalloproteins, while about 20% of the brain Zn^2+^ is in its free ionic form, presenting in the presynaptic vesicles of glutamatergic nerve terminals [20]. When glutamatergic neurons are excited, Zn^2+^ within vesicles is released into synaptic clefts along with glutamate [8], activating many receptors such as NMDA (N-methyl-d-aspartate), GABAA (γ-aminobutyric acid type A), AMPA (α-amino-3-hydroxy-5-methyl-4-isoxazole propionic acid) receptors, glycine receptors and voltage-gated ion channels, thus influences synaptic plasticity and transmission [21,22,23]. Studies suggest that excitatory cortico-amygdala synapses need Zn^2+^ to induce long-term potentiation (LTP) and form auditory fear memory [24]. Also, Zn^2+^ is necessary for the LTP induction of mossy fibers for information storage [25]. Significantly, Zn^2+^ is involved in somatosensory processing and precision perception of stimuli [15,26]. Glutamatergic Zn^2+^-enriched neurons (ZENs) are present in the dorsal cochlear nucleus and olfactory bulb, indicating that Zn^2+^ is involved in auditory and olfactory stimuli response [27].

Zn^2+^ is a crucial factor for several steps of neurogenesis including proliferation, migration, differentiation, and survival in both developmental and adult stages. Zn^2+^ deficiency results in impaired neuronal proliferation, differentiation, and activation of apoptotic pathways [20]. Indeed, evidence shows that the hippocampus is probably the most susceptible region to Zn^2+^ deficiency [18,28], which reduces progenitor cell number and neuronal differentiation, thus causing irreversible impairment of learning and memory capacity during early development [29,30].

As a redox-inert metal, Zn^2+^ exhibits strong antioxidant properties by a series of molecules and enzymes, thus playing a crucial role in resisting oxidative stress [31]. However, excess Zn^2+^ can cause neurotoxicity mainly through oxidative stress generation and is a risk factor for stroke, epilepsy, and ischemia [32]. Studies have shown that under detrimental stimuli, Zn^2+^ is released in large quantities from presynaptic terminals and enters postsynaptic neurons [33,34]. Significantly increased intracellular free Zn^2+^ enhances the level of neuronal NADPH oxidase subunit, which is involved in the generation of reactive oxygen species (ROS) [35,36].

Therefore, as a special class of neurotransmitter that cannot be metabolized or synthesized, Zn^2+^ needs to be tightly regulated to prevent neuronal death associated with Zn^2+^ dysregulation. In the brain, Zn^2+^ homeostasis is mainly maintained by three families of proteins, metallothioneins (MT), Zn^2+^ transporters (ZnT), and Zrt-, Irt-like proteins (ZIP) [14]. They work together to modulate Zn^2+^ signaling spatially and temporally.

MT is the major binding protein to buffer cytoplasmic Zn^2+^. It is a cysteine-rich peptide consisting of 61 amino acids. The 20 cysteine residues of MT are the binding sites of several divalent metals including Zn^2+^, copper (Cu^2+^), mercury (Hg^2+^) and cadmium (Cd^3+^) [11]. Each MT can bind to 7 Zn^2+^ atoms by metal-thiolate clusters as a reservoir [37]. The expression of MTs is induced by a high level of Zn^2+^. There are four isotypes of MTs, of which MT-1 and MT-2 are generally present in all cells. MT-3 is found mainly in the CNS, especially the glutamatergic neurons containing Zn^2+^, and MT-4 mainly exists in stratified epithelial cells [38,39]. MTs regulate and detoxify the intracellular heavy metals and are involved in metal absorption through the intestinal mucosa. They have specific roles in the immune system and transcription [40]. Evidence suggests that MTs have redox properties, exerting a critical role in the protection against oxidative stress [41].

ZnTs regulate intracellular Zn^2+^ by efflux and transport into intracellular compartments from the cytosol. There are currently 10 members of the ZnT family (ZnT1–ZnT10), which are designated SLC30. At present, most structural information about Zn^2+^ transporters is obtained from analysis and prediction studies on their prokaryotic homologs. Based on the crystal structure of their Escherichia coli homolog YiiP, ZnTs are predicted to have six transmembrane domains with cytosolic NH2 and COOH termini [42]. The expression of ZnTs is closely related to the fluctuations in Zn^2+^ concentration. Except for ZnT1, ZnTs are located in intracellular compartments and act as Zn^2+^/H+ antiporters. ZnT1 is found on the cell membrane of neurons and glia, transporting Zn^2+^ out of the cells, thus it is implicated in protection from neurotoxic Zn^2+^ surges during pathological conditions [43]. The expression of ZnT1 is upregulated by the increase of intracellular Zn^2+^ triggered by nitric oxide during cerebral ischemia. ZnT3 is the neuron-specific isoform found on synaptic vesicles and the only route by which Zn^2+^ can be transported into vesicles [36]. As the most extensively researched ZnT, ZnT3 is significant for brain Zn^2+^-related research. ZnT3-knockout (KO) mice lack synaptic Zn^2+^, and thus are a vital model for studying its role in pathological conditions of CNS. ZnT3 KO mice showed age-dependent cognitive impairment and increased susceptibility to seizures [44,45]. In addition, ZnT3 mRNA levels have been proven to decrease with age and in postmortem brain tissues of Alzheimer’s patients [46,47]. ZnT2, ZnT4, ZnT5, ZnT6 and ZnT10 are located in several regions of the brain. Among them, ZnT2, ZnT4 and ZnT10 transport Zn^2+^ into lysosomes, endosomes, and secretory vesicles, while ZnT5 and ZnT6 are involved in the transport of Zn^2+^ into Golgi apparatus and trans-Golgi network [48]. ZnT9 is expressed in the nucleus and cytoplasm of the cerebellum [49].

In contrast to ZnTs, ZIPs encoded by SLC39 genes transport Zn^2+^ from extracellular space and intracellular compartments into the cytoplasm to increase cytoplasmic Zn^2+^ level [50]. ZIP family constitutes 14 members (ZIP1–ZIP14), which are widely distributed in brain and peripheral tissues. ZIP transporters have been predicted to have eight transmembrane domains with extracellular NH2 and COOH termini [51]. Evidence suggests that the expression and location of ZIPs can also be regulated by the Zn^2+^ level [52]. Zn^2+^. Except for ZIP5, the expression of cell surface ZIPs generally increases under Zn^2+^ deficiency [5]. In addition to Zn^2+^, ZIPs can also transport other metals including iron (Fe^2+^), Cu^2+^, Cd^3+^ and manganese (Mn^2+^) [5,14]. Defects of ZIPs have been shown to be associated with several neurological diseases. ZIP8 mutations have been linked to cerebellar atrophy syndrome [53]. ZIP12, abundantly expressed in the brain, plays a crucial role in neurite outgrowth and tubulin polymerization [5].

## 3. Zn^2+^ and CNS Diseases

A growing body of evidence suggests that disruption of Zn^2+^ homeostasis is associated with the pathogenesis of a variety of CNS disorders, including Alzheimer’s disease, depression, Parkinson’s disease, multiple sclerosis, schizophrenia, epilepsy and traumatic brain injury. In this review, the roles of Zn^2+^ in these disorders were introduced. The main preclinical and clinical findings about the involvement of Zn^2+^ in several central nervous system disorders are summarized in Table 1.

### 3.1. Zn^2+^ and Alzheimer’s Disease

As a multifactorial chronic neurodegenerative disorder, Alzheimer’s disease (AD) is the most common cause of dementia, accounting for more than half of dementia cases. Age is considered the main risk factor for sporadic cases, which account for 95% of all AD cases. While other cases mostly have a genetic inheritance, called familial Alzheimer’s disease (FAD). FAD can be caused by mutations of several genes including presenilin 1 (PSEN1), presenilin 2 (PSEN2) and amyloid precursor protein (APP) [84]. AD is pathologically characterized by the deposition of β-amyloid (Aβ) plaques and hyperphosphorylated tau proteins (p-tau). Currently, the primary pathogenic factor of AD is still controversial, and the overall failure of drug discovery based on the Aβ and tau hypothesis implies that a reappraisal of the AD model is urgent. Given the critical neurological effects and the rich content of Zn^2+^ in the neocortex, the research on the role of Zn^2+^ in the pathogenesis of AD has advanced rapidly since Burnet et al. first described Zn^2+^ as a potential pathogenic factor of dementia [85].

Of all the studies on Zn^2+^ in the pathogenesis of AD, the involvement of Zn^2+^ in the deposition of Aβ was the most extensively researched. The imbalance between Aβ production and clearance is thought to be a likely drive of AD, in which Zn^2+^ plays a significant role [86]. Aβ, a peptide composed of 39–43 amino acids, is the cleavage product of amyloid precursor protein (APP) catalyzed by β-secretase and γ-secretase in sequence. Zn^2+^ has been shown to be involved in the synthesis and processing of APP to affect the production of Aβ. Some Zn^2+^-containing transcription factors such as p53 and NF-κB participate in the APP synthesis [87,88]. Zn^2+^ also modulates the expression levels and activity of α, β and γ-secretase. A high concentration of Zn^2+^ enhances the expression levels of β and γ-secretase while decreasing that of α-secretase [89]. Several Zn^2+^-binding sites have been found on APP, by which Zn^2+^ can influence the digestive efficiency of the 3 secretases through conformational changes [90]. There is a complex interplay between Zn^2+^ and Aβ. Zn^2+^ can interact with Aβ to regulate the polymerization of Aβ into different forms [91]. At low concentrations, by selective precipitation of aggregation intermediates, Zn^2+^ performs a protective effect on resisting the neurotoxicity of Aβ. While at high concentrations, Zn^2+^ binding increases the fibrillar Aβ aggregation [92]. Results from both in vitro and in vivo studies have shown that Zn^2+^-amyloid interactions promote the precipitation of Aβ fibrils, in which Zn^2+^ accumulates in large quantities [93,94]. Lee et al. have demonstrated that transgenic Tg2576 mice (mice overexpressing human APP) crossed with ZnT3 KO mice had markedly reduced plaque load, providing further evidence that Zn^2+^ contributes to Aβ aggregation [95]. By sequestrating Zn^2+^, Aβ depletes the Zn^2+^-mediated glutamatergic neurotransmission, leading to the neurotoxicity correlated with glutamatergic overdrive [96]. Metabotropic ZnR (GPR39) and TrkB receptors are also affected, which in turn perturb LTP [97]. Accumulated Aβ can incorporate into membranes to form unregulated amyloid channels, initiating a continuous inward flow of Ca^2+^ [98] (Figure 2). The consequent disruption of calcium homeostasis might be the primary cause of Aβ neurotoxicity [99].

Accumulating evidence implies that enhanced Aβ:metal interactions aggravate oxidative stress by uncontrolled production of ROS [100]. Specifically, the abundant Fe^3+^ and Cu^2+^ in the vicinity of Aβ provide the fuel for the generation of H2O2 and downstream ROS by Fenton and Haber-Weiss chemistry, affecting lipid peroxidation and the formation of DNA and protein adducts [101]. The conformational changes of Aβ induced by Zn^2+^ may exert a protective role by suppressing the interaction of oxidizing metals with Aβ and the ROS generation [102].

In addition, Zn^2+^ is involved in the clearance of Aβ. The sequestration of Zn^2+^ by Aβ affects the catalytic activity of several Zn^2+^ metalloproteinases which can degrade Aβ, including matrix metalloproteinases (MMP), neprilysin (NEP) and insulin-degrading enzyme (IDE) [97] (Figure 2). The Aβ-Zn^2+^ complex has increased resistance to proteolysis [103], and S100A6 from astrocytes can promote its clearance by sequestrating Zn^2+^ [104]. Whereas Zn^2+^ can also activate microglia to enhance the phagocytosis of Aβ [105] (Figure 2).

Neurotrophic signaling is also defective in AD. Zn^2+^ is critical to the modulation of neurotrophic signaling in AD, especially the brain-derived neurotrophic factor (BDNF)-TrkB axis [106]. BDNF level is associated with the severity of AD-related cognitive decline [107,108]. Zn^2+^-dependent MMPs (mainly MMP-2 and 9) are involved in the transformation of pro-BDNF to mBDNF [89] (Figure 2). It was observed that Zn^2+^ supplementation counteracted the decline in BDNF level through MMP activation in 3xTg-AD mice [56]. Pro-BDNF was found to inhibit GABAergic transmission [109], and this along with the impaired maturation of BDNF may provide a mechanism for neuronal death and altered neuronal excitability [96,110].

Aβ pathology is regarded as a biomarker of Zn^2+^ dyshomeostasis [111]. Due to the excessive interaction between Zn^2+^ and Aβ, extracellular and intracellular Zn^2+^ homeostasis are both impaired [89] (Figure 2). There may be 2 stages of Zn^2+^ dyshomeostasis in AD [97]: Slow turnover of synaptic Zn^2+^ in the early stage is vital for the formation of Aβ deposition. Reuptake of released Zn^2+^ after the glutamatergic transmission is an energy-dependent process, thus decreased mitochondrial energy during aging leads to increased average extracellular Zn^2+^ level, promoting Aβ aggregation [112]. Increased intracellular Zn^2+^ can only be observed in advanced AD, possibly because of the inhibitory effect on Zn^2+^ export of 4-hydroxynonenal, which is a peroxidation product generated by Aβ-Cu^2+^ complexes and elevated in AD [113].

Hyperphosphorylation of tau and the formation of neurofibrillary tangles (NFT) is another contributing factor to neuronal disorder in AD. It has been found that Zn^2+^ triggers tau phosphorylation by activation of glycogen synthase kinase 3β (GSK3β), cyclin-dependent kinase 5 (CDK5), extracellular signal-regulated protein kinase 1/2 (ERK1/2) and c-Jun N-terminal kinase (JNK) [111]. Recent evidence suggests that Zn^2+^ can inhibit protein phosphatase 2A (PP2A) via the Src-dependent pathway, leading to the hyperphosphorylation of tau [114] (Figure 2). In addition to phosphorylation, it was observed that Zn^2+^ can promote tau aggregation by direct interaction with tau [115]. Huang et al. have demonstrated that Zn^2+^ binding can also directly enhance tau toxicity using a drosophila tauopathy model [116]. The hyperphosphorylation of tau may be associated with the elevated Zn^2+^ level during advanced AD [97].

Many studies showed that Zn^2+^ homeostasis can affect Aβ/tau pathology and is the potential therapeutic target. Based on these findings, there are two main strategies for Zn^2+^-targeting therapy, including Zn^2+^ supplementation and modulation [89]. The role of Zn^2+^ supplementation in AD is not fully understood. Nutritional Zn^2+^ deficiency commonly occurred in the advanced age of AD. Low dietary Zn^2+^ has been found to increase the volume of amyloid plaques in APP/PS1 mice [117]. Zn^2+^ supplementation in 3xTg-AD mice was described to significantly delay memory deficits and reduce both Aβ and tau pathology [56]. On the contrary, it was found that Zn^2+^-enriched diets could impair spatial memory in Tg2576 and TgCRND8 transgenic models [118]. Also, in addition to spatial learning and memory impairment, Zn^2+^ treatment increased Aβ deposition in APP/PS1 mice [119]. Deficits in biochemistry and behavior triggered by tau were observed to be exacerbated by Zn^2+^ supplementation in a tau mouse model [120]. A meta-analysis indicated no evidence that Zn^2+^ supplementation can lower the risk of AD [57].

Zn^2+^ ionophores including clioquinol (CQ, 5-chloro-7-iodoquinolin-8-ol) and PBT2 (5,7-dichloro-2-[(dimethylamino)methyl]quinolin-8-ol) have shown positive results in Aβ reduction and cognition promotion in clinical trials [121,122]. These two metal-protein-attenuating compounds (MPACs) have been shown to promote the uptake of Zn^2+^ into cells with an impact on several relevant neurochemical pathways, thus normalizing the distribution of Zn^2+^ and Zn^2+^-dependent signaling pathways [123,124]. These findings further illustrated the critical role of Zn^2+^ homeostasis in AD and the complex interaction between them (Figure 2).

### 3.2. Zn^2+^ and Depression

Depression is a common mental disorder associated with high morbidity and mortality. A growing body of evidence suggesting that there is a correlation between Zn^2+^ and depression has sparked enormous interest. The link between Zn^2+^ deficiency and depression or depression-like symptoms has been observed in both animal models and clinical trials. Dietary Zn^2+^ deficiency was found to cause depression-like behaviors in rats [125]. Serum Zn^2+^ levels are markedly lower in major depressed patients compared to normal controls and there is a negative correlation between serum Zn^2+^ and the severity of depression [58]. It was reported that treatment-resistant depressed patients have lower Zn^2+^ levels than treatment-non-resistant patients [126]. A systemic review showed that Zn^2+^ significantly lowered depressive symptom scores of patients suffering from depression as an adjunct to antidepressant drug treatment [59]. Another recent meta-analysis revealed the role of Zn^2+^ supplementation in the improvement of depression status as a monotherapy [60]. In addition, postpartum Zn^2+^ supplementation was demonstrated to markedly improve maternal blood Zn^2+^ status with a positive effect on reducing the risk of postpartum depression [61].

At present, the underlying neurobiological mechanism of the association between depression and low Zn^2+^ levels is still unclear, and many pathways may be involved. Zn^2+^ has been shown to be implicated in abnormal endocrine pathways in the pathogenesis of depression. It was reported that the hypothalamic-pituitary–adrenal (HPA) system is disrupted in about 50% of depressed patients [127,128,129]. The increase in serum corticosterone has been demonstrated in Zn^2+^-deficient mice and rats [130,131]. It may be that the chronically increased serum glucocorticoid rather than insufficient chelatable Zn^2+^ is associated with the increase in depression-like behavior [132]. Repeated corticosterone injections were also found to increase depression-like behavior in rats and mice [133,134]. Increased glucocorticoid under Zn^2+^ deficiency may be involved in the change in the excitability of glutamatergic neurons. The excessive excitation of glutamatergic neurons in the hippocampus observed in Zn^2+^-deficient rats after exposure to acute stress may be due to aberrant glucocorticoid secretion, and contribute to stress susceptibility and increased depression-like behavior [135]. It is believed that corticosterone secretion during Zn^2+^ deficiency may be involved in the blocking of glutamate transporter, causing excess glutamate accumulation [136].

Zn^2+^ may exert antidepressant action through neurotransmission. Chronic treatment with citalopram and fluoxetine, which are selective serotonin reuptake inhibitors (SSRI), can significantly enhance the serum Zn^2+^ level [137,138]. The antidepressant-like effect of Zn^2+^ in the forced swim test (FST) was demonstrated to be markedly abolished by serotonin synthesis inhibitors or serotonin receptor antagonists [139]. The density of 5-HT1A and 5-HT2A receptors was enhanced by chronic Zn^2+^ administration in the hippocampus and frontal cortex, respectively [140]. The anti-immobility effect of Zn^2+^ was partially abolished in 5-HT1A autoreceptor KO mice [141]. Relevant studies have shown that Zn^2+^ can regulate serotonin signaling by the allosteric modulation of 5-HT receptors [141,142,143]. Zn^2+^ potentiates agonist binding to 5-HT1A receptor at sub-micromolar concentrations (10 μM) while exerting an inhibitory effect at sub-millimolar concentrations (500 μM) [141]. Additionally, the binding of the antagonist to the 5-HT1A receptor is inhibited by Zn^2+^ [142]. The oligomerization state between GalR1-5-HT1A-GPR39 is regulated by Zn^2+^ concentration and this may affect depressive behavior [144]. Zn^2+^ has been shown to disrupt the heterodimerization of 5-HT1A and GalR1 [145], which may be a novel therapeutic target of depression [146].

Previous studies have also shown that glutamatergic signaling is crucial in the pathology and treatment of depression. In the limbic system, Zn^2+^ typically acts as an inhibitory modulator of the NMDA receptor [147]. *N*-methyl-d-aspartic acid (NMDA) administration was shown to antagonize the antidepressant-like activity of Zn^2+^ [148]. Coadministration of ineffectively low doses of NMDA antagonists and ineffective doses of Zn^2+^ became effective in FST [148]. A NMDA receptor agonist was observed to block the antidepressant-like effect of both Zn^2+^ and magnesium (Mg) [149]. The affinity of glycine to glycine/NMDA receptors was demonstrated to be decreased by chronic Zn^2+^ administration [140]. Increased levels of GluN2A and GluN2B in the rat hippocampus were found following Zn^2+^ restriction [150]. Also, the role of the AMPA receptor in antidepressant-like effects of Zn^2+^ has been shown. An antagonist of the AMPA receptor abolished the antidepressant-like activity of Zn^2+^ while an AMPA receptor potentiator presented a synergistic effect with Zn^2+^ in the FST [148].

Emerging evidence also suggests the involvement of metabotropic mGluRs in depression. Several antagonists of group I and group II mGluRs have exhibited antidepressant-like effects in the FST and mouse tail suspension test (TST) [151,152]. Zn^2+^ has been demonstrated to be an antagonist of group I and group II mGluRs (Figure 3), and this is a potential mechanism of antidepressant-like effects of Zn^2+^ [153].

Zn^2+^ may act as a neurotransmitter through another metabotropic receptor, GPR39 [154]. GPR39 KO mice exhibited depressive-like behavior in FST and TST [155]. A decreased level of GPR39 was observed in the hippocampus and frontal cortex of Zn^2+^-deficient rats and mice [156]. Suicide victims also show a lower level of hippocampal and cortical GPR39 [156]. In the study by Omar and Tash, a joint administration of fluoxetine and Zn^2+^ significantly increased the hippocampal protein level of GPR39 [157]. There are data suggesting that GPR39 is also involved in glutamatergic neurotransmission. Zn^2+^ was found to upregulate the activity of potassium chloride co-transporter 2 (KCC2) by a soluble *N*-ethylmaleimide-sensitive factor attachment protein receptor (SNARE)-dependent process via GPR39 [158] (Figure 3). KCC2 is a transporter that plays a crucial role in the maintenance of neuronal chloride gradient, enabling the generation of inhibitory currents within postsynaptic neurons through GABAA receptors [159]. By the enhanced activity of KCC2, Zn^2+^ can provide protection from high-level stimulation and resultant excitotoxicity [159]. The activation of GPR39 by Zn^2+^ is needed for the synthesis of endocannabinoid 2-arachidonoylglycerol (2-AG) (Figure 3), which exerts an inhibitory effect on glutamate release [160]. Notably, monoamine-based antidepressants had no effectiveness in GPR39 KO mice, indicating that GPR39 is indispensable for the effects of drugs targeting the 5-HT system [161].

Neurotrophic signaling also plays an indispensable role in depression. Two meta-analyses showed that BDNF level decreases in depressed subjects and BDNF level markedly increases after antidepressant treatment [162]. Peripheral BDNF administration produced antidepressant-like responses at both behavioral and cellular levels and increased hippocampal neurogenesis [163]. BDNF may be regarded as a crucial biomarker of major depressive disorder (MDD) [164]. Chronic treatment with Zn^2+^ was observed to increase BDNF levels in the hippocampus and cortex in rats [165,166], implying the potential involvement of Zn^2+^ in the depression via BDNF system. Zn^2+^ has been demonstrated to transactivate TrkB and its downstream signaling pathways, including extracellular signal-regulated kinase (ERK1/2), cAMP response element binding protein (CREB), and phospholipase C-γ (PLC-γ) by increasing the activity of Src family kinase independent of BDNF [167,168] (Figure 3). This finding is in accordance with increased cortical ERK phosphorylation after Zn^2+^ treatment in a study by Franco et al. [166]. In addition, data suggest that the BDNF pathway depends on the GPR39. In GPR39 KO mice, the CREB and BDNF levels were decreased in the hippocampus [155]. Acute administration of TC-G 1008 (a GPR39 agonist) was found to increase hippocampal GPR39 and BDNF levels [169]. Activation of GPR39 seems to cause CREB over-expression via cAMP responsive element (CRE)-mediated transcription and thus up-regulates the BDNF expression [136]. Another potential mechanism involving CREB/BDNF pathway may be the antagonism of glycogen synthase kinase-3 (GSK-3). Zn^2+^ can suppress the phosphorylation activity of GSK-3, which inhibits the activity of CREB and is negatively regulated by BDNF [153] (Figure 3). Thus, Zn^2+^ can enhance BDNF function via GSK-3 inhibition.

A growing body of evidence also suggests the involvement of neuronal precursor cells in Zn^2+^ deficiency-induced depression. About a 50% decrease in proliferating cells was observed in the subgranular zone (SGZ) and granular cell layer of the dentate gyrus in the rats treated with a Zn^2+^-restricted diet [170]. Similar findings were also demonstrated in Zn^2+^-deficient mice [171]. There was also a significant increase in the number of TUNEL-labeled cells in the SGZ, which is relevant to p53-dependent apoptotic mechanisms [170]. On the other hand, the number of dividing cells in the hilus and dentate gyrus of the hippocampus was reported to markedly increase by chronic antidepressant treatment [172]. Many imaging studies have identified reductions in hippocampal volume in patients with depression [173]. In addition to proliferation, Zn^2+^ deficiency also impairs neuronal differentiation, and this may be partially mediated by changes in TGF-β signaling [174].

The antioxidative activity of Zn^2+^ may be involved in its antidepressant effects. Increased ROS and lipid peroxidation and decreased superoxide dismutase (SOD) levels were found in depressed patients [173,175]. Eight weeks of antidepressant treatment has been shown to increase SOD activity and decrease nitric oxide (NO) levels [175]. There is evidence showing that the antidepressant properties of Zn^2+^ are related to the involvement of the L-arginine-nitric oxide (NO) pathway, in which Zn^2+^ acts as an inhibitor of nitric oxide synthase (NOS) [176]. Malathion is a toxic organophosphate that could cause depressant-like behavior and oxidative damage in rodents [177,178], while the depressant-like effect and a series of oxidative damage caused by malathion were attenuated by Zn^2+^ [177]. The total glutathione level in the hippocampus and cerebral cortex was significantly increased accompanied by an antidepressant-like effect by a chronic Zn^2+^ treatment in rats, indicating an enhancement of antioxidant buffering capacity over time [166]. It was reported that Zn^2+^ supplementation reversed lithium-induced reductions in catalase and glutathione-s-transferase (GST) activities in the cerebellum, relieving the oxidative stress caused by lithium toxicity [179].

Extensive evidence suggests that MDD is accompanied by the activation of an inflammatory response with changes in inflammatory markers [180]. Increased inflammatory activity was found in rat models of depression treated by bilateral olfactory bulbectomy [181] or chronic mild stress (CMS) [182]. Depressed patients have been shown to have an increased number of monocytes, neutrophils and T-lymphocytes and increased generation of proinflammatory cytokines including IL-1, IL-6 and TNF-α [183,184]. Additionally, the indoleamine 2,3-dioxygenase (IDO) activated by proinflammatory cytokines metabolizes tryptophan into quinolinic acid, which is a NMDA receptor agonist involved in depression [147]. An acute phase (AP) response with alterations in AP proteins is also involved in depression, while one characteristic of acute phase response is reduced serum Zn^2+^ level [153]. It is thought that the Zn^2+^ reduction is due to Zn^2+^ sequestration by up-regulation of metallothionein in the liver, bone marrow and thymus, which is induced by increased proinflammatory cytokines [185,186]. Furthermore, IL-6-induced increase in Zip14 also plays a role in the mechanism of hypozincemia in acute phase response [187]. A significant negative correlation between serum IL-6 and Zn^2+^ levels was also reported [188]. In addition, the concentration of albumin, the major Zn^2+^ binding protein in serum, is significantly reduced in MDD patients and there is a markedly positive correlation between serum albumin and Zn^2+^ level [189,190], indicating the potential role of the carrier protein of Zn^2+^ in hypozincemia during depression. These findings indicate that inflammatory response is linked to a crucial Zn^2+^ redistribution [180].

Besides, there is a negative correlation between neopterin and Zn^2+^ levels, suggesting that hypozincemia is also associated with the activation of cell-mediated immunity, which is related to increased cytokines such as IL-6 [58]. Zn^2+^ plays a significant role in cell-mediated immunity, and the availability of Zn^2+^ is indispensable for the normal functions of T cells and B cells [180,191]. Low Zn^2+^ level was shown to be correlated with a decrease in the CD4+/CD8+ T cell ratio [191,192]. Cytokines secreted by type 1 helper T (Th1) cells including IFN-γ and IL-2 are decreased under Zn^2+^ deficiency, suggesting an impairment of Th1 function [193,194], while one function of IFN-γ is to inhibit the development of the Th17 cells [180], which play a significant role in autoimmune diseases. The study by Chen et al. described an imbalance of Th17/Treg ratio in MDD patients [195]. Zn^2+^ was also reported to inhibit Th17 development through attenuation of STAT3 activation [196]. The suppression of the proliferation and IL-17 production of stimulated human T cells by zinc aspartate provides further evidence of the relationship between Zn^2+^ and Th17 cells [68]. In addition, a significantly increased subset of B cells producing autoantibodies was found in MDD patients [197]. These findings suggest that there is an impaired cell-mediated immune function accompanied by a tendency towards autoimmune activity in depression [195].

### 3.3. Zn^2+^ and Parkinson’s Disease

As the second most common neurodegenerative disorder after AD, Parkinson’s disease (PD) is a long-term CNS disorder affecting the motor system. The symptoms of PD progress slowly over time, with the most common and usually the first symptom of PD being bradykinesia, which is followed by other characteristic symptoms including tremors, muscle rigidity, postural instability and hypokinesia [198]. In addition to motor symptoms, non-motor symptoms such as psychiatric symptoms, fatigue, impaired sense of smell, sleep problems and symptoms associated with autonomous systems are also common in PD [199]. At present, the etiology of PD is not fully understood and 75% of cases are known as idiopathic PD, besides, more than 90% of PD cases are sporadic, whereas genetic factor only accounts for 5–10% of cases [198]. A wealth of evidence has implied the potential involvement of Zn^2+^ in the pathogenesis of PD. Multiple studies found a decreased circulating Zn^2+^ level in PD patients [200,201,202,203], whereas there is evidence showing a normal or even increased Zn^2+^ level [204]. Furthermore, recent meta-analysis studies showed that the Zn^2+^ levels in serum, plasma, and cerebrospinal fluid (CSF) are reduced in PD patients [62,63,64]. In a drosophila model with a mutant Parkin, which is involved in familiar PD, Zn^2+^ supplementation improved both lifespan and motor abilities [65]. Quiroga et al. reported a case of PD with low Zn^2+^ and vitamin C levels whose movement disorder was rapidly resolved after the replacement of Zn^2+^ and vitamin C, highlighting the correlation between PD and Zn^2+^ deficiency [205].

It is thought that the Zn^2+^ deficiency found in PD patients is attributed to the antioxidative properties of Zn^2+^ [206]. Oxidative stress is thought to play a crucial role in the pathogenesis of PD [207]. Many markers of oxidative stress, such as nucleic acid oxidation [208], lipid peroxidation [209] and protein nitration [210] have been observed in dopaminergic brain regions. In rats with PD induced by rotenone, Zn^2+^ supplementation decreased lipid peroxidation and cell death, indicating that the neuroprotective role of Zn^2+^ may be mediated by its antioxidant effects [211]. Loss of ATP13A2 (PARK9), a lysosomal type 5 P-type ATPase which is correlated with early-onset PD, was shown to lead to Zn^2+^ dyshomeostasis, impaired mitochondrial function and altered ROS metabolism [212], providing further evidence of the association between Zn^2+^ and oxidative stress in PD.

PD is pathologically characterized by the loss of dopaminergic neurons within substantia nigra pars compacta, subsequently leading to decreased dopamine secretion. Evidence suggests that excessive Zn^2+^ is linked to dopaminergic neurodegeneration. Post-mortem studies showed Zn^2+^ depositions in the substantia nigra of idiopathic PD patients [206]. In mouse models of PD induced with MPTP, the accumulation of Zn^2+^ was observed in degenerating dopaminergic neurons [213]. It was also demonstrated that the Zn^2+^ level increased in all structures located along the dopaminergic pathway in the 6-OHDA-induced parkinsonian brain [214]. In addition, Zn^2+^ chelation attenuated the neuronal death induced by MPP+, which is used to model PD [215]. These findings collectively suggest that the cytosolic labile accumulation of Zn^2+^ contributes to the death of dopaminergic neurons in PD.

Another characteristic feature of PD is the Lewy body, an insoluble aggregate composed of α-synuclein. It was reported that the accumulation of α-synuclein in dopaminergic neurons caused apoptosis, which required endogenous production of dopamine and was mediated by ROS, while non-dopaminergic neurons were not affected, providing a possible explanation for the selective loss of dopaminergic neurons in PD [216]. The increase of α-synuclein expression by METH was found to be reversed by pretreatment with Zn^2+^Cl2 [217]. Additionally, intracellular Zn^2+^ dyshomeostasis induced by PARK9 loss of function was demonstrated to cause accumulation of α-synuclein [218].

### 3.4. Zn^2+^ and Multiple Sclerosis

Multiple sclerosis (MS) is a chronic autoimmune disorder of CNS characterized by immune cell infiltration and myeline sheath damage. T cells with autoimmune activity can become sensitive to endogenous myelin to initiate the demyelinating process of the myelinated structures, leading to impairment of the communication of the brain and remaining body parts, as a result, MS patients suffer from coordination difficulties, muscle weakness and even irreversible neurological damage [219].

Currently, the exact etiology of MS is not clear, it is thought that several factors including infection, genetics, immunology, and environment contribute to the disorder [220]. Accumulating evidence suggests that the disruption of Zn^2+^ homeostasis is associated with the pathogenesis of MS. Several studies have shown that MS patients have both decreased serum Zn^2+^ levels [202,221,222] and reduced Zn^2+^ intake [223], while others showed no significant differences in serum Zn^2+^ level between healthy controls and MS patients [224], or even higher plasma Zn^2+^ concentration in MS patients [225]. However, two recent meta-analysis studies revealed lower circulating Zn^2+^ levels in MS [66,67]. It was reported that erythrocyte Zn^2+^ levels in patients with active disease exhibited a significant decrease during the early stages of exacerbation and a gradual increase with the recovery from the attack [226]. These findings suggest the potential role of Zn^2+^ dyshomeostasis during MS.

In terms of the mechanism, abnormal synaptic release and intracellular accumulation of Zn^2+^ are believed to be involved in multiple steps of MS including the activation of MMP-9 and disruption of the blood-brain barrier (BBB) and subsequent infiltration of immune cells [227]. Clioquinol (CQ) and ZnT3 gene deletion were shown to significantly suppress EAE (an animal model of MS)-associated clinical features and neuropathological changes, as well as inhibit MMP-9 activation, BBB disruption and immune cell infiltration, implying the involvement of synaptic Zn^2+^ in myelin damage of spinal cord white matter [227,228]. Likewise, similar results were also found in a recent study using 1H10, a novel Zn^2+^ chelator [229].

Besides, activation of NADPH mediated by Zn^2+^ plays a role in the microglial activation and oligodendrocyte death, which act as mediators in the MS. Evidence suggests that Zn^2+^ alone can induce NADPH oxidase activation and subsequent ROS generation, which have been demonstrated to significantly contribute to the MS pathogenesis [230]. Apocynin, an NADPH oxidase assembly inhibitor, was reported to decrease clinical symptoms of EAE and MOG-induced proliferation, morphology transformation and pro-inflammatory cytokine release of cultured microglia [230]. In terms of oligodendrocyte death, the peroxynitrite produced through NADPH and iNOS is thought to be a candidate. Peroxynitrite was reported to promote Zn^2+^ release from intracellular stores, resulting in sequential activation of ERK1/2 and 12-lipoxygenase (12-LOX), ROS production and eventual oligodendroglia death. Meanwhile, toxicity induced by peroxynitrite could be blocked completely by *N*,*N*,*N*′,*N*′-tetrakis (2-pyridylmethyl)ethylenediamine (TPEN), a Zn^2+^ chelator. In addition, it was observed that AMPA-mediated Ca^2+^-dependent Zn^2+^ accumulation participates in the excitotoxic injury of oligodendrocytes. The mechanism underlying this may be associated with Zn^2+^-induced ROS production, ERK1/2 and PARP-1 activation, leading to cell death [231].

However, zinc aspartate was shown to inhibit proliferation and cytokine production in stimulated human T cells and mouse splenocytes in vitro, and reduce the clinical severity of EAE [68,69]. Furthermore, high concentrations of Zn^2+^ were observed to impair the metabolic fitness and differentiation of Th1 cells, a significant participant of MS, and decrease Th1 autoimmune inflammation [232]. Based on present findings, it seems to be paradoxical that both Zn^2+^ excess and deficiency are correlated with MS. One interpretation of this is the specific local Zn^2+^ requirements, to be specific, MS is a complex disease accompanied by both demyelination and remyelination processes, although Zn^2+^ excess has been shown to cause white matter damage, the following remyelination phase may require Zn^2+^ influx to proceed [233]. Current studies underline the complex interrelationship between Zn^2+^ and MS and provide novel insight into Zn^2+^ as a potential therapeutic target of MS, while the related molecular mechanisms need to be further investigated.

### 3.5. Zn^2+^ and Schizophrenia

Schizophrenia (SCZ) is a severe psychotic disorder with both neurodevelopmental and degenerative pathologies affecting approximately 1% of the population, and its typical symptoms can be divided into positive, negative, and cognitive symptoms [147,234]. The underlying causes of the SCZ phenotype involve the interactions of prenatal Zn^2+^ deficiency with genetic risk factors [147]. Moreover, fetal Zn^2+^ deficiency has been shown to be linked to maternal exposure to infectious agents, and evidence suggests that maternal infection is a key risk factor for the development of SCZ in offspring [74]. Given the crucial role of Zn^2+^ in the immune system and neurodevelopment, prenatal Zn^2+^ deficiency associated with inflammation causes deleterious effects in the fetal brain. Maternal immune activation induced by LPS can be utilized as a model of SCZ, and prenatal Zn^2+^ supplementation was observed to attenuate the behavioral impairments [74], as well as alleviate the increased pro-inflammatory mediator expression, microglial and astrocyte density in the prefrontal cortex (PFC) of male offspring prenatally exposed to LPS [235].

Besides, compared with the control group, a 30–50% decrease in brain Zn^2+^ content was found in postmortem brain tissue of early onset SCZ patients [14]. Several studies have revealed reduced serum Zn^2+^ levels in SCZ patients [236,237,238]. Two recent meta-analysis studies also showed significantly lower serum and blood Zn^2+^ concentrations in SCZ patients [70,71]. The lower Zn^2+^ level in SCZ can lead to NMDA hyperactivity and possibly psychotic symptoms [147]. It was demonstrated that a combination of zinc sulfate and risperidone was effective in the improvement of SCZ symptoms and aggression risk, indicating the potential of Zn^2+^ as an adjuvant agent of SCZ management [72]. Another study further highlighted the possibility of Zn^2+^ as a therapeutic agent for SCZ management [73]. These findings illustrate the participation of Zn^2+^ dyshomeostasis in the pathogenesis of SCZ.

In addition, abnormalities of intracellular Zn^2+^ due to the dysfunction of molecules transporting Zn^2+^ are also involved in SCZ. One missense SNP of SLC39A8 (ZIP8 gene), rs13107325, was reported to be strongly associated with SCZ [239]. Whereas the potential mechanism by which this missense variant confers the risk of SCZ is not clear. Using a knock-in mouse model, in which a threonine was introduced at the 393rd amino acid of mouse SLC39A8 to correspond to rs13107325 (p.Ala391Thr) of human SLC39A8, Li et al. demonstrated dysregulation of Zn^2+^ level in the blood and brain and significantly reduced cortical dendritic spine density in the SLC39A8-p.393T knock-in mice, providing a reasonable explanation of the biological effects of rs13107325 in SCZ, as it is believed that dysfunction of dendritic spines may play a critical role in SCZ [240]. Furthermore, it was reported that this missense variant resulted in increased innate immune response and glutamate receptor hypofunction, which was partly associated with decreased surface expression of glutamate receptor subunits, further illustrating the role of rs13107325 [241]. In addition, mRNA for SLC39A12 (ZIP12 gene) was found to increase in the dorsolateral prefrontal cortex of SCZ patients [242]. Emerging data suggest that polymorphisms of SLC30A3, the gene encoding ZnT3, is also linked to a higher risk of schizophrenia [243]. Of note, genetic associations are emerging in other gene families related to Zn^2+^, such as the ZNF family, in which Zn^2+^ acts as a structural cofactor [244]. Among this family, ZNF804A is a typical example, and it has been demonstrated to be correlated with several psychiatric disorders including SCZ [245]. Current research provides novel insight into the involvement of Zn^2+^ and possible interventions in SCZ, while it is obvious that a better understanding is needed.

### 3.6. Zn^2+^ and Epilepsy

Epilepsy is a common neurological disorder affecting 65 million people worldwide [246]. It is characterized by recurrent and spontaneous seizures. Currently, antiepileptic drug (AED) therapy is the treatment of choice for epilepsy, although most patients can attain remission using AED, more than one-third of patients have resistance [247]. The available drugs can relieve symptoms rather than improve the underlying state or progression [248]. Therefore, the development of novel medication is still urgent. An imbalance between neuronal excitation and inhibition has been established as a potential explanation of epilepsy [249], however, the mechanisms underlying epilepsy development and progression are still unclear, further limiting the search for novel drugs. Zn^2+^ is able to exert effects that either increase or decrease neuronal excitability, implying the possibility of Zn^2+^ involvement in the pathogenesis of epilepsy.

Although there is evidence from both preclinical and clinical studies corroborating the association between Zn^2+^ and epilepsy, the results are conflicting. Several studies reported a decreased serum Zn^2+^ level in children with epilepsy [75,76,250]. However, a meta-analysis found a significantly higher serum Zn^2+^ concentration in nontreated patients with epilepsy [77]. Also, this study revealed a significantly lower serum Zn^2+^ level in epileptic patients receiving valproate monotherapy compared to epileptic patients without anticonvulsant therapy, which was consistent with the findings of another study reporting a markedly decreased serum concentration in epileptic patients after valproic acid treatment [251]. Moreover, a medium dose of Zn^2+^ has been demonstrated to reduce the severity of pilocarpine-induced limbic seizures either as a monotherapy or in combination with valproic acid in a rat model of epilepsy, while a high dose of Zn^2+^ exacerbated that, confirmed a dose-dependent effect of Zn^2+^ [78]. A double-blind, placebo-controlled trial found that 31% of children with refractory epilepsy who received Zn^2+^ treatment for six months showed significant improvement, compared with only 4.5% of patients treated with a placebo [79]. These findings suggest the critical role of Zn^2+^ in epilepsy and the potential add-on effect of Zn^2+^ supplementation in AED treatment.

The association between Zn^2+^ and epilepsy is complex and poorly understood. Evidence suggests that inflammation may act as both a consequence and the cause of epileptic seizures. Brain inflammation can be caused by seizures, and chronic inflammation can be aggravated by recurring seizures [80]. Studies have demonstrated that neuroinflammation with increased pro-inflammatory cytokines, such as IL-1β, lead to seizure-evoked excitotoxicity [252,253]. It was reported that consumption of antioxidants [254], mitochondrial dysfunction and activated intrinsic pathway components [255] may also be involved in epilepsy. Furthermore, the study by Baraka et al. [78] showed that an appropriate dose of Zn^2+^ can mitigate seizures by suppressing increased IL-1β levels, oxidative stress, and apoptotic activity, which was consistent with another recent study by Mehmet et al. [80], providing further evidence.

The involvement of proteins regulating Zn^2+^ homeostasis were also reported in epilepsy. Using animals lacking genes encoding specific proteins regulating Zn^2+^ homeostasis is a useful method to clarify the role of these proteins in epilepsy. Qian et al. [256] showed increased susceptibility to seizures induced by kainic acid (KA) in mice lacking ZnT3. Likewise, a lower seizure threshold to KA was found in ZIP-1,3-null mutants. In addition, there was a significant difference in lethality caused by seizures, with 17% for mutants compared to 4% for the control group following a 15 mg/kg KA injection. Also, this study found that the decrease of Zn^2+^ uptake either by knockout of ZIP-1,3 transporters or abolishment of synaptic Zn^2+^ by knockout of ZnT3 exerts a protective role in neurodegeneration after prolonged seizures [256].

In conclusion, as the pathways linking Zn^2+^ and the regulation of epilepsy are not fully understood, further research is needed to assess the efficacy of Zn^2+^-based therapy for epilepsy.

### 3.7. Zn^2+^ and Traumatic Brain Injury

Traumatic brain injury (TBI) is a cerebral pathology that has become a severe public health and socioeconomic issue in the modern world. It affects 64–79 million people worldwide annually [257]. TBI patients can develop many impairments in communication, sensory processing, and cognition. In addition, behavioral problems such as anxiety and depression can also be found in TBI patients [258].

The role of Zn^2+^ in TBI has been studied over the past three decades. Emerging evidence suggests that Zn^2+^ seems to have both neurotoxic and neuroprotective roles in TBI [258]. Thus, there have been many studies examining the potential benefits of Zn^2+^ chelation or Zn^2+^ supplementation.

Following TBI, excessive Zn^2+^ releases from presynaptic vesicles acutely, and the inappropriate accumulation of intracellular Zn^2+^ in postsynaptic neurons leads to excitotoxicity and cell death [259]. Therefore, the possible application of Zn^2+^ chelators has been investigated. Calcium disodium ethylenediaminetetraacetate (CaEDTA) has been demonstrated to upregulate several neuroprotective genes and reduce apoptotic cell death after TBI [260]. Zhao et al. [261] reported that Zn^2+^ translocation and autophagy were inhibited by TPEN following TBI.

Although Zn^2+^ exerts a deadly role in acute neuronal damage after injury, it is crucial for longer-term reparative processes including neurogenesis and gene expression [262]. In a study by McClain et al. [81], decreased serum Zn^2+^ levels and increased urine Zn^2+^ excretion which was proportional to the severity of injury were found. In addition, moderate Zn^2+^ deficiency has been observed to increase cell death after TBI [263]. Furthermore, a recent multicenter prospective study [264] reported an association between serum Zn^2+^ deficiency and a higher incidence of disability and long- and short-term mortality in TBI patients with intracranial injury. On the other hand, it was shown that although Zn^2+^ chelation could provide short-term histological neuroprotection, it failed to improve neurobehavioral outcomes, including spatial memory deficits, after TBI [265]. Another study revealed significantly decreased BrdU, Ki67 and DCX positive cells following CQ treatment, indicating that neurogenesis induced by TBI is suppressed by Zn^2+^ chelation [266], further stressing the potential role of free Zn^2+^ in the repair of TBI.

As a result, there has been a recent trend to investigate the efficacy of appropriately timed Zn^2+^ supplementation in the improvement of outcomes following TBI. Evidence suggests that Zn^2+^ supplementation is effective in improving cognitive impairment and depression in animal models [82]. Also, it was shown that Zn^2+^ supplementation before injury could provide more protection from depression and anxiety associated with TBI than using Zn^2+^ solely as a treatment, indicating the possible use of chronic Zn^2+^ supplementation for populations at risk for TBI [82]. Similarly, in a double-blinded controlled trial, 68 patients with severe brain injury were randomly assigned to either a standard Zn^2+^ group or Zn^2+^-supplemented group. One month after injury, the standard Zn^2+^ group had a mortality rate of 26%, compared with only 12% in the Zn^2+^-supplemented group [83].

Although the Zn^2+^ supplementation has been demonstrated to be effective in improving outcomes associated with TBI in current studies, future research is needed to determine the optimal timing and dosage of Zn^2+^ intervention to provide behavioral resiliency to TBI as a treatment.

## 4. Conclusions

As described above, current findings have consolidated that Zn^2+^ exerts a critical role in the initiation and progress of the pathological features of multiple CNS disorders. Zn^2+^ dyshomeostasis acts as an intersection of the pathogenesis and symptoms of these disorders. Therefore, the maintenance of Zn^2+^ homeostasis is crucial as any substantial alterations may lead to deleterious outcomes. As the acceleration of the appreciation for the relevance of Zn^2+^ in the pathogenesis of CNS disorders, restoration of Zn^2+^ homeostasis has been regarded as a potential therapeutic target. Although Zn^2+^ supplementation has shown some beneficial effects in the improvement of symptoms and pathological characteristics, considering the neurotoxicity of excess Zn^2+^ and the heterogeneity of neurological disorders, the dose of Zn^2+^ supplementation should be under strict control. Zn^2+^ chelation is another significant aspect of the intervention of CNS disorders, especially AD, nevertheless it also shows some side effects due to the chelation of other essential divalent metal ions, highlighting the timing of intervention. Thus, understanding how Zn^2+^ level changes over the course of the disorders is the key to optimize a Zn^2+^-based targeted therapy. In conclusion, a better comprehension of the underlying mechanisms linking Zn^2+^ and these CNS disorders is urgent and may contribute to the acceptance of Zn^2+^ as a key point in the CNS disorders. We hope that this review can provide new clues for the prevention and treatment of nervous system disorders.

## Figures and Tables

**Figure 1 nutrients-15-02140-f001:**
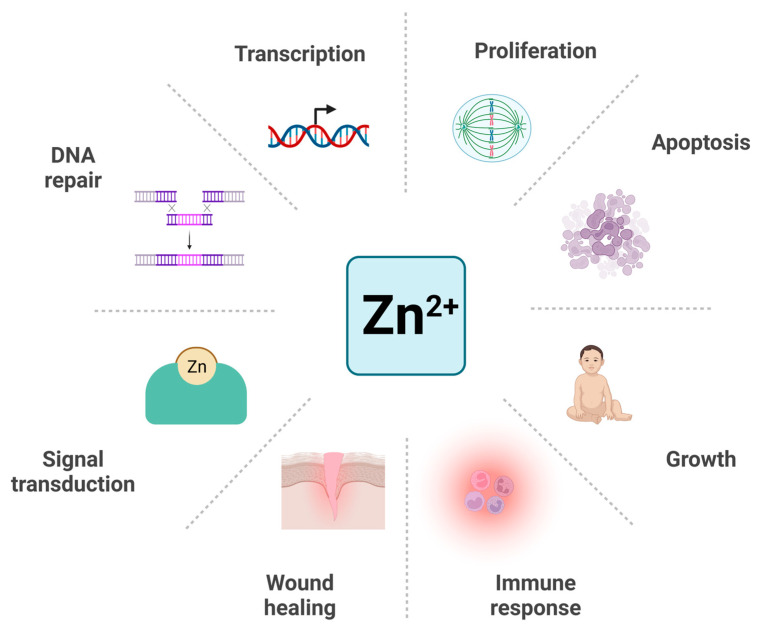
The involvement of Zn^2+^ in the physiological processes. Zn^2+^ participates in many physiological processes including DNA repair, transcription, proliferation, apoptosis, growth, immune response, wound healing, and signal transduction. Created with BioRender.com. Adapted from “Circular Diagram with 8 Sections (Layout)”, by BioRender.com (2023). Retrieved from https://app.biorender.com/biorender-templates (accessed on 5 February 2023).

**Figure 2 nutrients-15-02140-f002:**
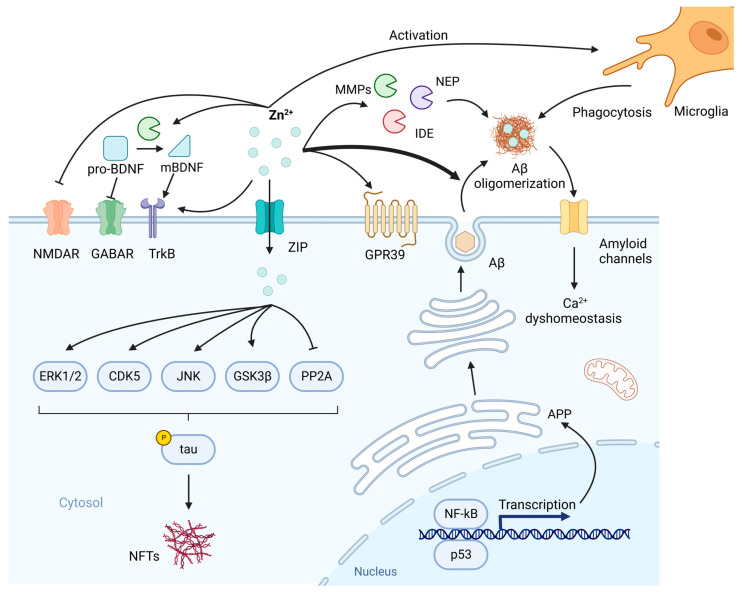
The involvement of Zn^2+^ in the pathogenesis of Alzheimer’s disease (AD). Zn^2+^-containing transcription factors NF-κB and p53 participate in the synthesis of amyloid precursor protein (APP) and subsequent β-amyloid (Aβ). After production, Aβ is secreted upon neuronal activity and aggregated by Zn^2+^. In normal conditions, Zn^2+^ inhibits *N*-methyl-d-aspartate receptor (NMDAR) to regulate synaptic neurotransmission. Also, Zn^2+^ is involved in the normal memory events such as long-term potentiation (LTP) by TrkB and GPR39. brain-derived neurotrophic factor (BDNF)-TrkB axis needs to be modulated by Zn^2+^ because Zn^2+^-dependent MMPs are indispensable to the transformation of pro-BDNF to mBDNF, while pro-BDNF inhibits GABAergic transmission. In addition, the clearance of Aβ is affected by Zn^2+^. Zn^2+^ metalloproteinases including matrix metalloproteinases (MMPs), neprilysin (NEP) and insulin-degrading enzyme (IDE) are involved in the degradation of Aβ, while Zn^2+^-loaded Aβ oligomers have increased resistance to proteolysis. Microglia can be activated by Zn^2+^ to enhance the phagocytosis of Aβ. Due to the sequestration of Zn^2+^ by Aβ, these normal neurobiological activities in which Zn^2+^ plays a crucial role are disrupted. Aβ can incorporate into membranes to form amyloid channels, as a result, an inward flow of Ca^2+^ is initiated, leading to Ca^2+^ dyshomeostasis. On the other hand, intracellular Zn^2+^ can lead to phosphorylation of tau by activating extracellular signal-regulated protein kinase 1/2 (ERK1/2), cyclin-dependent kinase 5 (CDK5), c-Jun N-terminal kinase (JNK), glycogen synthase kinase 3β (GSK3β) and inhibiting protein phosphatase 2A (PP2A). Created with BioRender.com. Adapted from “Drosophila Toll Pathway”, by BioRender.com (2023). Retrieved from https://app.biorender.com/biorender-templates (accessed on 9 February 2023).

**Figure 3 nutrients-15-02140-f003:**
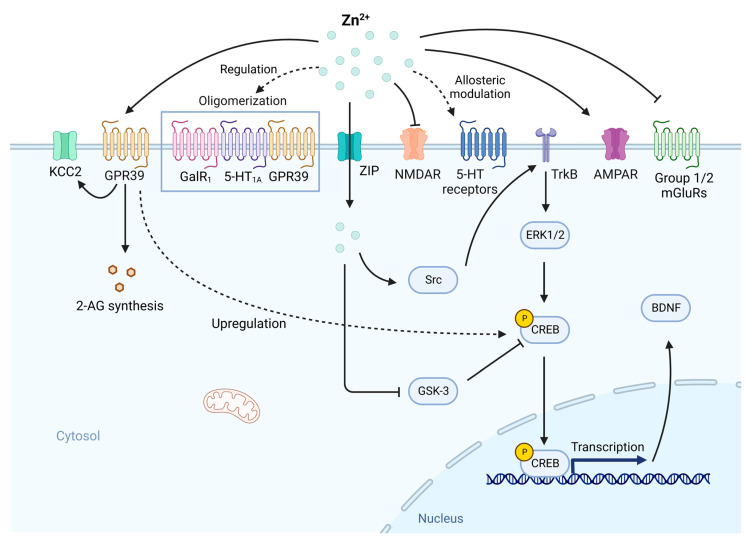
The involvement of Zn^2+^ deficiency in the abnormal neurotransmission and neurotropic signaling in depression. In normal conditions, the activities of glutamatergic receptors including NMDAR and α-amino-3-hydroxy-5-methyl-4-isoxazole propionic acid receptor (AMPAR) need to be regulated by Zn^2+^. The activation of GPR39 by Zn^2+^ is essential for endocannabinoid 2-arachidonoylglycerol (2-AG) synthesis and the upregulation of potassium chloride co-transporter 2 (KCC2). GPR39 activation can upregulate cAMP response element binding protein (CREB) expression, thus increase BDNF expression. Zn^2+^ is involved in the allosteric modulation of 5-HT receptors and regulation of the oligomerization state between GalR1-5-HTA1-GPR39. Zn^2+^ is also an antagonist of groupⅠand group Ⅱ mGluRs, which are linked to depression. While intracellular Zn^2+^ participates in the modulation of neurotropic signaling, which is crucial in depression. Zn^2+^ can transactivate TrkB and downstream ERK1/2 and CREB via Src family kinase, on the other hand, inhibit glycogen synthase kinase-3 (GSK-3), which inhibits CREB activity, leading to the expression of BDNF. Under Zn^2+^ deficiency, these activities associated with depression are disrupted, contributing to the development of depression. Created with BioRender.com. Adapted from “Drosophila Toll Pathway”, by BioRender.com (2023). Retrieved from https://app.biorender.com/biorender-templates (accessed on 17 February 2023).

**Table 1 nutrients-15-02140-t001:** Summary of the main preclinical and clinical findings about the involvement of Zn^2+^ in several central nervous system disorders.

Disorder	Serum/Plasma Zn^2+^ Status	Zn^2+^ Supplementation
Alzheimer’s disease (AD)	↓ [54,55]	Delay memory deficits and reduce both β-amyloid (Aβ) and tau in 3xTg-AD mice [56]
		No conclusive evidence [57]
Depression	↓ [58]	Lower depressive symptom scores as an adjunct [59]
		Improve depression status as a monotherapy [60]
		Postpartum Zn^2+^ supplementation reduced the risk of postpartum depression [61]
Parkinson’s disease (PD)	↓ [62,63,64]	Improve both lifespan and motor abilities in a drosophila model with a mutant Parkin [65]
Multiple sclerosis (MS)	↓ [66,67]	Reduce clinical signs in animal models [68,69]
Schizophrenia (SCZ)	↓ [70,71]	Effective as an adjuvant agent [72]
		Effective as a therapeutic agent [73]
		Prenatal Zn^2+^ supplementation attenuated the behavioral impairments [74]
Epilepsy	↓ [75,76] ↑ [77]	A medium dose of Zn^2+^ reduced the severity of pilocarpine-induced limbic seizures either as a monotherapy or in combination with valproic acid in a rat model of epilepsy [78]
		Reduce seizure frequency in children with refractory epilepsy [79]
		Mitigate seizures by alleviating inflammation and oxidative stress [78,80]
Traumatic brain injury (TBI)	↓ [81]	Effective in improving cognitive impairment and depression in animal models [82]
		Reduce mortality rate in a double-blinded controlled trial [83]

## Data Availability

No new data were created or analyzed in this study. Data sharing is not applicable to this article.

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
