# Peer review of "Zinc and Central Nervous System Disorders"

_nutrients, 2023, doi:10.3390/nu15092140_

Round 1
Reviewer 1 Report
This review presents the latest data on the role of zinc in some disorders of the central nervous system. The review contains quite comprehensive information, useful for the field. The whole manuscript is also well written.
Detailed comments can be found below.
Please cite Figures in the text.
Introduction section: I agree, it is clearly written, but not complete, missing an important message - the novelty of this review.
This topic has been previously reviewed, so it would have been better to stress how this review differs from previous ones and what is the new contribution to the field?
Lines 87-91 - references needed.
Line 132 - reference to "kidney and colon cells" - excluded from the aim of the review.
Sentence Line 138 "ZIP12, abundantly expressed in the brain, plays a crucial role in neurite growth and tubulin polymerisation". - needs references.
Lines 251-255 need references.
Lines 262-263 need references.
Besides depression, AD, PD, MS and SCD, there are other aspects of zinc involvement in central nervous system disorders that should be included, such as epilepsy, traumatic brain injury and stroke.
In the chapter "Zn in the CNS" the authors discuss Zn transporters in detail. It would be better to add some more information in the other chapters on zinc transporters and their involvement in brain diseases.
Reviewer 2 Report
In the present review, B. Wang and H. Chen have considered the role of zinc in defined diseases of the central nervous system (Alzheimer´s Disease, Depression, Parkinson´s Disease, Multiple Sklerosis and Schizophrenia). The topic of the review is very interesting and up-to-date. The review is clearly structured and the included tables and figures are very informative.
However, I have some comments that I would like to address:
- At the end of page 6, line 263 a reference for the meta-analysis is missing.
- Page 9, line 374: change “neurotropic” in “neurotrophic”.
- The preclinical findings for Multiple Sklerosis in the EAE model in mice (see page 13) should be included in Table 1.
